# Predictors of in-hospital mortality among cirrhotic patients in Ethiopia: A multicenter retrospective study

Tamrat Petros Elias[ID][1,2]*, Abate Bane Shewaye[1,3], Henok Fisseha[2],
Abdulsemed Mohammed Nur[1,3], Kaleb Assefa Berhane[1], Asteray Tsige Minyilshewa[1],
Kibrab Bulto Kumsa[2], Biruck Mohammed Seid[3]

**1** Department of Internal Medicine, Adera Medical and Surgical Center, Addis Ababa, Ethiopia,
**2** Department of Internal Medicine, St. Paul's Hospital Millennium Medical College, Addis Ababa, Ethiopia,
**3** Department of Internal Medicine, College of Health Science, Addis Ababa University, Addis Ababa,
Ethiopia

* tamrat.petros@sphmmc.edu.et

## Abstract

### Background

In Ethiopia, cirrhosis is the 6th leading cause of death and is responsible for high hospitalization and mortality rates. However, until now, factors affecting in-hospital mortality of patients with liver cirrhosis are poorly understood. This study assessed the predictors of in-hospital mortality among cirrhotic patients in Ethiopia.

### Methods

A retrospective cross-sectional study using data collected from the electronic medical records of patients who were admitted for complications of liver cirrhosis between January 1, 2023, and March 31, 2024, in the medical wards of Adera Medical Center, St. Paul's Hospital Millennium Medical College, and Tikur Anbessa Specialized Hospital. Frequency and cross-tabulation were used for descriptive statistics. Predictor variables with a p-value <0.25 in bivariate analyses were included in the logistic regression.

### Results

Of the 299 patients included in the final analysis, the majority (79.6%) were males, and the median age of the study participants was 45 (IQR, 36–56) years. Hepatitis B virus (32.1%) was the most common etiology, followed by alcohol (30.1%) and hepatitis C virus (13.4%). Ascites (69.2%), upper gastrointestinal bleeding (50.5%), and hepatic encephalopathy (44.8%) were the most common forms of presentation. The in-hospital mortality rate was 25.4%. West Haven grade III or IV hepatic encephalopathy (AOR: 12.0; 95% CI 2.33–61.63; P<0.01), hepatocellular carcinoma (AOR: 9.05;

**Editor:** Alessandro Granito, University Hospital of Bologna Sant'Orsola-Malpighi Polyclinic Department of Digestive System: Azienda Ospedaliero-Universitaria di Bologna Policlinico Sant'Orsola-Malpighi Dipartimento dell'apparato digerente, ITALY

**Data availability statement:** All relevant data are within the manuscript.

**Funding:** The author(s) received no specific funding for this work.

**Competing interests:** The authors have declared that no competing interests exist.

95% CI 2.18–37.14; P: 0.01), history of previous admission within one year period (AOR: 6.80; 95% CI 2.18–21.18; P < 0.01), acute kidney injury (AOR: 6.47; 95% CI 1.77–23.64; P < 0.01), and model for end-stage liver disease – sodium score (AOR: 1.17; 95% CI 1.05–1.30; P: 0.02), were found to be predictors of in-hospital mortality.

## Conclusion

In-hospital mortality of cirrhotic patients is high in Ethiopia. West Haven grade III or IV hepatic encephalopathy is the leading cause of mortality. Hence, prompt identification and management of hepatic encephalopathy and its precipitant at an earlier stage is crucial for better treatment outcomes and survival.

## Introduction

Cirrhosis is the fibrotic replacement of liver tissue that can result from any chronic liver disease (CLD) [1]. It is a leading cause of liver-related death worldwide. Globally, the estimated number of deaths associated with cirrhosis in 2019 was 1.47 million accounting for 2.4% of all-cause mortality. This number increased by 10% after 2010 [2]. According to the global burden of disease super-regions for 2017, the age-standardized death rate was highest in sub-Saharan Africa [3]. In Ethiopia, cirrhosis is the 6th leading cause of mortality, accounting for 24 deaths/100,000 population [4].

In the initial stages, cirrhosis is compensated and patients are asymptomatic. Decompensation is usually defined as the first occurrence of ascites, esophageal variceal bleeding, or hepatic encephalopathy [3]. The transition from a compensated state to a decompensated state occurs at a rate of 5–12% per year [2]. Once decompensation occurs, the mortality and morbidity resulting from cirrhosis also increase sharply, and the 1-year case-fatality rate can reach 80%. Quality of life is also affected and frequent hospitalizations (admissions and stays) are needed [3]. When patients with cirrhosis of the liver are hospitalized, mortality increases significantly, ranging from 44 to 74% [5].

The major etiologies of cirrhosis are hepatitis B virus (HBV) and hepatitis C virus (HCV) infection, alcohol-associated liver disease, metabolic dysfunction-associated steatotic liver disease (MASLD), and autoimmune hepatitis. Globally, among individuals with cirrhosis, 42% had HBV infection and 21% had HCV infection [2]. In sub-Saharan Africa, 69% of liver cirrhosis cases are attributed to HBV, HCV, and alcohol consumption [6]. HBV (40%), alcohol (17%), and HCV (15%) are the three most common etiologies of CLD in Ethiopia [7].

In Ethiopia, liver disease accounts for 12% of admissions and 31% of hospital mortality in adult medical wards [8]. On the other hand, the overall hospital mortality rate in CLD patients ranges from 25–41% [7,9]. However, the factors affecting in-hospital mortality in admitted liver cirrhosis patients are poorly understood in many regions of Africa including Ethiopia [3]. The few studies conducted in Africa were limited by small sample sizes [10–13]. This study aimed to assess predictors

of in-hospital mortality among cirrhotic patients in Ethiopia on a relatively larger sample size. This manuscript has been prepared and reported in accordance with the Strengthening the Reporting of Observational Studies in Epidemiology (STROBE) guidelines [14]. The STROBE checklist has been completed and included as a supplementary file.

## Methodology

### Study design

A retrospective cross-sectional study design was used to assess the predictors of in-hospital mortality among patients who were admitted with a diagnosis of liver cirrhosis between January 1, 2023, and March 31, 2024, in the medical wards of Adera Medical Center, St. Paul's Hospital Millennium Medical College, and Tikur Anbessa Specialized Hospital.

### Study area

The study was conducted at Adera Medical Center, St. Paul's Hospital Millennium Medical College, and Tikur Anbessa Specialized Hospital. Adera Medical Center is a renowned private health facility in Addis Ababa that provides dedicated gastroenterology and hepatology services for patients from all over the country. St. Paul's Hospital Millennium Medical College is a tertiary-level teaching hospital under the Federal Ministry of Health with more than 700 beds in inpatient capacity. Tikur Anbessa Specialized Hospital is the largest referral university hospital in the country with more than 800 beds.

### Study participants

All patients ≥ 18 years of age admitted to the medical wards of Adera Medical Center, St. Paul's Hospital Millennium Medical College, and Tikur Anbessa Specialized Hospital with the diagnosis of liver cirrhosis between January 1, 2023, and March 31, 2024, were included in the study. Exclusion criteria were admission for non-cirrhosis-related causes like infection except spontaneous bacterial peritonitis, heart failure, trauma, etc., multiple (≥ 20%) missing variables on the patient charts, admission for observation after prophylactic endoscopic variceal ligation or percutaneous liver biopsy, and if the outcome of treatment is unknown due to discontinuation of treatment against medical advice or if it is not documented.

### Data collection tools and procedures

Sociodemographic (age and sex), clinical (etiology, duration since diagnosis, comorbidity, major signs and symptoms at clinical presentation, history of previous admission within one year, admission diagnosis, Child-Pugh score which assesses liver disease severity using five parameters: encephalopathy, ascites, bilirubin, albumin, and prothrombin time, to classify patients into classes A, B, or C, and model for end-stage liver disease – sodium (MELD-Na) score which predicts 90-day mortality in liver disease patients, incorporating bilirubin, INR, creatinine, and sodium levels at admission, duration of hospital stay, and outcome of hospital admission), laboratory (complete blood count, liver and renal function tests, and electrolytes), and imaging (fibroscan, abdominal ultrasound, computed tomography (CT) scan, and upper gastrointestinal endoscopy) data were extracted from the electronic medical records system between May 1, 2024, and May 31, 2024, by general practitioners working in the health facilities by using a pretested data abstraction format that was prepared by reviewing relevant works of literature. The data quality was ensured through training of the data collectors, close supervision, and prompt feedback. The data were checked for inconsistencies, completeness, accuracy, clarity, coding errors, and missing values, and appropriate corrections were made by the principal investigator.

### Sample size and statistical analysis

All patients with liver cirrhosis admitted to the medical wards of the three health facilities during the study period were included in the study. A total of 356 patients were admitted to the medical wards of the three hospitals with a diagnosis of

liver cirrhosis: 236 patients at Adera Medical Center, 71 patients at St. Paul's Hospital Millennium Medical College, and 49 patients at Tikur Anbessa Specialized Hospital. Of the 356 patients, 331 patients fulfilled the inclusion criteria, and 32 patients met the exclusion criteria; of which, 7 were due to multiple missing variables, 4 were due to unknown treatment outcome, 12 were due to admissions for observation post prophylactic endoscopic variceal ligation or percutaneous liver biopsy, and 9 were due to admissions for non-cirrhosis related causes. Hence, 299 patients were included in the final analysis.

Statistical Package for Social Sciences (SPSS) version 26 was used to enter and analyze the data. Frequency and cross-tabulation were used to summarize the descriptive statistics of the data. Associations between predictor variables and outcomes of interest were estimated using both bivariate analysis and binary logistic regression. The chi-square test and t-test were used in the bivariate analysis. Predictor variables with a p-value <0.25 in bivariate analyses were reported and included in the logistic regression. For the binary logistic regression, a 95% confidence interval for adjusted odds ratio (AOR) was calculated and variables with p-value<0.05 were considered statistically significant.

### Operational definition

*Liver cirrhosis* was diagnosed based on the presence of two or more of the following [1]:

1. Clinical signs of cirrhosis (jaundice, ascites, caput medusa, clubbing, palmar erythema, spider naevi, gynecomastia, female pubic hair pattern, encephalopathy, splenomegaly, or asterixis)

2. Impaired liver function test consistent with cirrhosis (International Normalized Ratio (INR) ≥ 1.5 and serum albumin ≤ 3.4gm/dl)

3. Imaging diagnosis of cirrhosis (surface nodularity, coarse and heterogeneous echotexture/ attenuation, segmental atrophy or hypertrophy, ascites, splenomegaly, gall bladder wall thickening, or portal vein diameter >13mm on abdominal ultrasound and/or CT scan)

4. Transient elastography (FibroScan®) > 12.5 Kilopaskal (KPa)

5. AST (aspartate aminotransferase) to Platelet Ratio Index (APRI) score of ≥ 1.5 or Fibrosis-4 (FIB-4) score ≥ 3.25

### Ethical consideration

The study was conducted after obtaining ethical clearance from Adera Medical Center Institutional Review Board (IRB), Ref No: 087/24. The IRB of the center waived the need for obtaining informed consent as only anonymized data from participants was collected retrospectively from the electronic medical record system. Confidentiality of individual patient information was maintained by using code numbers instead of other identifiers and the information gained from the chart was used only for research purposes.

## Results

### Sociodemographic and clinical characteristics

The median age of the study participants was 45 (interquartile range (IQR), 36–56) years and the majority (79.6%) of patients were males. HBV (32.1%) was the most common etiology followed by alcohol (30.1%) and HCV (13.4%). The cause of cirrhosis was not identified in 12% of the patients. MASLD, autoimmune hepatitis, and Budd Chiari syndrome accounted for 7.4%, 6.4%, and 2.3% of the etiology respectively. More than half (52.9%) of the patients were in the Child-Pugh class C category, 35.2% were in class B, and 11.9% were in class A. Approximately one-quarter (26.1%) of the patients had comorbidities. Diabetes (17.1%) was the most common comorbidity followed by hypertension (9.7%). From the signs and symptoms at admission, ascites (69.2%) and fatigue (56.2%) were the most common. Half (50.5%) of the

patients presented with upper GI bleeding (UGIB). All patients who presented with UGIB were found to have bleeding esophageal varices on endoscopic evaluation. Hepatic encephalopathy was present in 44.8% of the patients. The encephalopathy was grade one (13.4%), two (55.2%), three (23.9%), and four (7.5%) based on the West Haven criteria. UGIB (27.6%) was the most common precipitant for encephalopathy followed by electrolyte imbalance (23.1%) and spontaneous bacterial peritonitis (SBP) (14.2%). One-fourth (25.8%) of the patients had acute kidney injury (AKI) and 24.8% of the patients had SBP. Hepatocellular carcinoma (HCC) was present in 18.4% of the study participants.

### Factors associated with in-hospital mortality

Of the 299 patients included in the study, 76 patients (25.4%) have passed away in the hospital. As shown in Table 1, factors associated with in-hospital mortality in the bivariate analysis were, age, jaundice, ascites, pedal edema, change in mentation, sleep disturbance, admission within one year, AKI, HCC, hepatic encephalopathy, MELD-Na score, and Child-Pugh score. From the laboratory parameters, leukocyte count, hemoglobin, AST (aspartate aminotransferase), ALT (alanine aminotransferase), ALP (alkaline phosphatase), bilirubin level, serum albumin, INR (international normalized ratio), creatinine, urea, and serum sodium level have a statistically significant correlation with in-hospital mortality (Table 1).

Multivariate analysis was performed by using binary logistic regression to identify associations between variables with a P-value less than 0.05 in the bivariate analysis and in-hospital mortality. A history of previous admission within one year period (AOR: 6.80; 95% CI 2.18–21.18; $P < 0.01$), Grade III or IV hepatic encephalopathy (AOR: 12.0; 95% CI 2.33–61.63; $P < 0.01$), AKI (AOR: 6.47; 95% CI 1.77–23.64; $P < 0.01$), HCC (AOR: 9.05; 95% CI 2.18–37.14; P: 0.01), and MELD-Na Score (AOR: 1.17; 95% CI 1.05–1.30; P: 0.02), were found to have a statistically significant association with in-hospital mortality (Table 2).

### Discussion

In Ethiopia, cirrhosis of the liver is the cause of close to one-third of deaths in adult medical wards [8]. However, the predictors of in-hospital mortality among cirrhotic patients have not been well-studied. This study aimed to assess these factors in two referral teaching hospitals and one medical center with specialized gastroenterology and hepatology services. A total of 356 cirrhotic patients were admitted to the three health facilities during the study period, of which, 299 patients fulfilled the inclusion criteria and were included in the final analysis. The majority (79.6%) of patients were males. The median age of the study participants was 45 (IQR, 36–56) years and Hepatitis B virus (32.1%) was the most common etiology. More than half (52.9%) of the patients were in the Child-Pugh class C category. The prevalence of in-hospital mortality was 25.4%. In the binary logistic regression, history of previous admission within one one-year period, Grade III or IV hepatic encephalopathy, AKI, HCC, and MELD-Na Score were found to be significantly associated with in-hospital mortality.

The median age of the patients admitted to the hospitals was 45 (IQR, 36–56) years. This relatively young age is concerning, as cirrhosis can significantly impact patients' quality of life and life expectancy. This can have major socioeconomic consequences and can lead to disability, lost productivity, and the need for significant medical care and support. A similar age group was reported in Ethiopia [9] and other African studies [11–13]. However, this age is lower than those found in previous US [15] and European [16] research. This could be because of the high viral hepatitis burden, acquisition of these viruses at an early age, and a limited access to quality healthcare and liver disease screening in many African settings which can lead to delayed diagnoses, allowing earlier stages of hepatitis to progress further before patients receive appropriate management.

The majority (79.6%) of the patients in this study were males. A higher prevalence of cirrhosis in males was reported in multiple other studies [17–19]. Men are more likely to develop liver cirrhosis and its complications due to a combination of behavioral and biological factors. Higher alcohol consumption is a key behavioral factor, while lower estrogen levels, which have anti-fibrotic and anti-inflammatory effects, increase their biological susceptibility to liver disease progression [20].

**Table 1. Bivariate analysis of factors associated with in-hospital mortality.**

| Characteristics | Discharged alive | Death | P-Value |
|---|---|---|---|
| Median age in years (IQR) | 45.0 (35.0-55.0) | 48.5 (39.25–57.25) | **0.02** |
| Gender | | | 0.62 |
| Male | 176 (73.9%) | 62 (26.1%) | |
| Female | 47 (77.0%) | 14 (23.0%) | |
| Etiology of cirrhosis | | | |
| Alcohol | 66 (73.3%) | 24 (26.7%) | 0.74 |
| Hepatitis B Virus | 72 (75.0%) | 24 (25.0%) | 0.90 |
| Hepatitis C Virus | 27 (67.5%) | 13 (32.5%) | 0.27 |
| MASLD | 20 (90.9%) | 2 (9.1%) | 0.06 |
| Autoimmune hepatitis | 15 (78.9%) | 4 (21.1%) | 0.65 |
| Budd-Chiari Syndrome | 5 (71.4%) | 2 (28.6%) | 0.84 |
| Unknown | 25 (69.4%) | 11 (30.6%) | 0.45 |
| Comorbidity | | | 0.14 |
| Yes | 63 (80.7%) | 15 (19.3%) | |
| No | 160 (72.4%) | 61 (27.6%) | |
| Specific comorbidity | | | |
| Diabetes | 44 (86.3%) | 7 (13.7%) | 0.08 |
| Hypertension | 23 (79.3%) | 6 (20.7%) | 0.53 |
| Asthma or COPD | 10 (90.9%) | 1 (9.1%) | 0.20 |
| HIV | 2 (50.0%) | 2 (50.0%) | 0.25 |
| Chronic kidney disease | 8 (72.7%) | 3 (27.3%) | 0.88 |
| Sign and Symptom | | | |
| Fatigue | 115 (68.5%) | 53 (31.5%) | 0.16 |
| Jaundice | 86 (60.6%) | 56 (39.4%) | **<0.01** |
| Ascites | 146 (70.5%) | 61 (29.5%) | **0.01** |
| Pedal edema | 66 (61.1%) | 42 (38.9%) | **<0.01** |
| Change in mentation | 49 (51.6%) | 46 (48.4%) | **<0.01** |
| Sleep disturbance | 50 (61.7%) | 31 (38.3%) | **0.04** |
| Melena or hematemesis | 123 (84.8%) | 22 (15.2%) | 0.53 |
| History of admission within one year | | | **<0.01** |
| Yes | 60 (60.6%) | 39 (39.4%) | |
| No | 163 (81.5%) | 37 (18.5%) | |
| Diagnosis at admission | | | |
| SBP | 52 (70.3%) | 22 (29.7%) | 0.32 |
| AKI | 33 (42.8%) | 44 (57.2%) | **<0.01** |
| Variceal UGIB | 127 (84.1%) | 24 (15.9%) | 0.78 |
| Hepatic encephalopathy | | | **<0.01** |
| Grade I or II | 66 (71.7%) | 26 (28.3%) | |
| Grade III or IV | 11 (26.2%) | 31 (73.8%) | |
| HCC | 33 (60.0%) | 22 (40.0%) | **<0.01** |
| Laboratory Parameters (Mean ± SD) | | | |
| Leukocyte count (× 10⁹/L) | 7.5±4.3 | 11.0±5.4 | **<0.01** |
| Hemoglobin (gm/dl) | 11.5±3.0 | 10.7±2.7 | **0.03** |
| AST (IU/L) | 129.9±58.5 | 279.7±68.9 | **<0.01** |
| ALT (IU/L) | 84.0±61.7 | 176.6±77.1 | **<0.01** |
| ALP (IU/L) | 231.9±71.6 | 392.9±86.5 | **<0.01** |

*(Continued)*

**Table 1.** (Continued)

| Characteristics | Discharged alive | Death | P-Value |
|---|---|---|---|
| Total Bilirubin (mg/dL) | 4.1±1.8 | 10.5±2.7 | **<0.01** |
| Serum albumin (gm/dL) | 2.8±0.7 | 2.5±0.6 | **0.01** |
| INR | 1.6±0.4 | 2.0±0.9 | **<0.01** |
| Creatinine (mg/dL) | 1.1±0.6 | 2.0±1.4 | **<0.01** |
| Serum Sodium (mEq/L) | 133.1±8.3 | 128.0±10.3 | **0.01** |
| Serum Potassium (mEq/L) | 3.9±0.7 | 4.4±0.9 | 0.47 |
| Mean duration of stay in days | 6.4±4.2 | 7.6±3.7 | 0.21 |
| Mean Child-Pugh Score | 9.0±2.2 | 11.9±1.9 | **<0.01** |
| Mean MELD-Na Score | 18.8±7.3 | 30.1±6.2 | **<0.01** |

IQR: Inter quartile range; MASLD: metabolic dysfunction-associated steatotic liver disease; COPD: chronic obstructive pulmonary disease; HIV: human immunodeficiency virus; SBP: spontaneous bacterial peritonitis; AKI: acute kidney injury; UGIB: upper gastrointestinal bleeding; HCC: hepatocellular carcinoma; AST: aspartate aminotransferase; ALT: alanine aminotransferase; ALP: alkaline phosphatase; INR: international normalized ratio; MELD-Na: model for end-stage liver disease – sodium

**Table 2. Crude and adjusted odds ratio of predictors of In-hospital mortality.**

| Characteristics | COR (95% CI) | P-Value | AOR (95% CI) | P-Value |
|---|---|---|---|---|
| Grade III or IV HE | 4.2 (3.0–5.8) | <0.01 | 12.0 (2.3–61.6) | <0.01 |
| HCC | 1.8 (1.2–2.7) | <0.01 | 9.0 (2.2–37.1) | 0.01 |
| Admission within one year | 2.1 (1.4–3.1) | <0.01 | 6.8 (2.2–21.2) | <0.01 |
| AKI | 3.9 (2.7–5.7) | <0.01 | 6.5 (1.7–23.6) | <0.01 |
| MELD-Na Score | – | – | 1.2 (1.1–1.3) | <0.01 |

COR: Crude odds ratio; AOR: Adjusted odds ratio; HE: hepatic encephalopathy; HCC: hepatocellular carcinoma; AKI: acute kidney injury; MELD-Na: model for end-stage liver disease – sodium.

In our study, the most common etiology was hepatitis B virus (32.1%) followed by alcohol (30.1%) and hepatitis C virus (13.4%). HBV was also reported as the leading cause of cirrhosis in other studies conducted in Ethiopia [7,9]. Similar results were observed in research done in Togo [21], Ghana [22], Nigeria [23], and other Sub-Saharan African countries [24]. HCV is the major cause of cirrhosis in the Eastern Mediterranean region [2,25] and North Africa [26,27]. Alcohol was the dominant etiology in reports from India [19,28], Thailand [29], and Colombia[17]. Although lower than the global average, the peculiar finding in our study, when compared to other similar studies in Ethiopia, is the increasing prevalence of alcohol related liver disease and MASLD. This can be linked to the increasing trends in hazardous alcohol consumption [30] and metabolic risk factors for MASLD such as diabetes [31] in Ethiopia. This was also evident in our study, which revealed that 17.1% of the patients had diabetes and 26.1% of the patients had comorbidities. This represents a substantial change from an earlier Ethiopian study that reported the prevalence of diabetes and comorbidities in general to be 6.4% and 11% respectively [10]. This finding aligns with the global pattern that indicates the impending eclipse of the influence of viral hepatitis by emergent metabolic CLDs [2].

Ascites (69.2%), UGIB (50.5%), and hepatic encephalopathy (44.8%) were the most common presentations in this study. The prevalence of ascites in the current study is comparable to the previous studies conducted in Ethiopia and Ghana [9,12]. However, the finding significantly differs from those of former studies conducted in Ethiopia which reported UGIB to be present in 10.2% and 25.7% of the cases [9,10]. The figure in our study is also higher than those reported by

researchers from the United States (8.6%), Madagascar (33.3%), and Colombia (17.3%) [11,15,17]. The primary reason for this is that patients are usually referred to these centers for therapeutic endoscopic interventions due to the restricted availability of these services at other hospitals. The most frequent precipitant of hepatic encephalopathy in our patients was likewise discovered to be UGIB, which may account for why our study's prevalence of hepatic encephalopathy was higher than that of previous investigations.

In this study, the in-hospital mortality rate was 25.4%. This is comparable with the 25.9% and 23.5% in-hospital mortality rates that have been reported from studies previously conducted in Madagascar and Colombia, respectively [11,16]. However, this percentage was less than that reported by studies carried out in Saudi Arabia (35%), Ghana (41.3%), and the Ivory Coast (42.2%) and much greater than that reported in Pakistan (15.7%), Morocco (8.7%), and the US (6.6%) [12,13,15,26,32,33]. These discrepancies may have resulted from variances in the study settings, as the study in Saudi Arabia included patients admitted to the ICU, and the baseline characteristics of the patients. These attributes include the stage of the disease, related comorbidities, specific complications of liver cirrhosis that resulted in hospitalization, and the clinical condition of the patients at admission. The higher mortality rate seen in the current study compared to that of the US, Pakistan, and Morocco can also be attributed to patients' delayed presentations and the lack of treatment options, such as shunt surgeries, which serve as a bridge to more definitive options such as liver transplantation nationwide.

Hepatic encephalopathy is one of the most common complications of liver cirrhosis and results in a spectrum of neuropsychiatric symptoms caused by circulating gut-derived toxins of nitrogenous compounds. The spectrum of hepatic encephalopathy ranges from minimal brain function deficits, known as minimal hepatic encephalopathy, to hepatic coma. Overt hepatic encephalopathy occurs in approximately 30%–45% of patients with cirrhosis, while minimal hepatic encephalopathy may affect up to 60% of patients with chronic liver disease and up to 80% with cirrhosis. One way of classifying hepatic encephalopathy is according to the severity of manifestations based on West Haven classification [34]. In our study, West Haven grade III or IV hepatic encephalopathy was found to be an independent predictor of in-hospital mortality. Multiple similar findings were reported from Ghana, Morocco, Madagascar, and the United States [11,26,34,35]. This underlines the need for prompt identification of hepatic encephalopathy and its precipitant at an earlier stage, followed by proper management.

History of previous admission within one year was 33.1% and was also found to be a predictor of in-hospital mortality in the logistic regression. A similar finding was reported in Spain and Canada [36,37]. One possible explanation for the greater death rate in our patients with a history of readmission could be the substantially greater prevalence of hepatic encephalopathy (59.6% vs. 37.5%) in this group of patients than in the patients without such a history. Hepatic encephalopathy was also found to increase readmission and mortality in a study conducted in Italy [38]. This may also be exacerbated by the unavailability of rifaximin in Ethiopia, which has been demonstrated to lower the risk of overt hepatic encephalopathy recurrence [39].

Patients with cirrhosis may have AKI for a variety of reasons. Some of these include hepatorenal syndrome (HRS), which is characterized by renal vasoconstriction secondary to splanchnic pooling of blood that reduces the effective circulating blood volume; decreased renal perfusion due to gastrointestinal bleeding; use of diuretics; diarrhea caused by the use of lactulose or infections; and so on [40]. Regardless of the etiology, AKI was associated with increased in-hospital mortality in our study, which is similar to the findings of studies in the United States and Turkey [41,42]. This was also demonstrated in another systematic review and meta-analysis [43]

Close monitoring, early intervention, and addressing the underlying causes of AKI in cirrhosis patients are crucial to improve patient outcomes and reduce the risk of mortality.

The MELD Na score was also found to be an independent predictor of in-hospital mortality in our study. Similar studies performed in Brazil, and Poland also showed that the MELD Na score predicts in-hospital mortality in cirrhotic patients [44,45]. Incorporating the MELD Na score into routine clinical practice can help identify high-risk cirrhosis patients and guide the management approach to improve patient outcomes and reduce mortality.

HCC was the other predictor of hospital mortality in this study. Similar findings were reported in other Sub-Saharan African countries such as Ivory Coast and Ghana [13,46]. However, the presence of HCC was not found to be a predictor of in-hospital mortality in research done in the US [15]. Because of poor screening and surveillance of HCC in cirrhotic patients, 95% of HCC cases in Sub-Saharan Africa are diagnosed late in the advanced or terminal stages, whereas 40% of cases in high-income countries are diagnosed at an early stage [47]. This, combined with the very limited availability of curative therapies, may have contributed to the results observed in our study. Improving early detection, enhancing access to therapies, and tailored management approaches for HCC could significantly improve patient outcomes and reduce the mortality associated with it.

UGIB was not shown to predict in-hospital mortality. The literature shows mixed results on the effect of UGIB on in-hospital mortality. A previous study done in Ethiopia [10] and another study conducted in Ghana [12] showed a significant association. In contrast, a study from France [48] showed a different result. Endoscopic therapy and antibiotic prophylaxis were shown to be independent predictors of survival in the French study. The reduced mortality found in our study could be due to the relatively better availability of interventional endoscopic services in the centers where our research was conducted, coupled with the current standard use of short-term antibiotic prophylaxis for SBP.

The major limitation of our study emanates from its retrospective design. The information collected from the electronic medical records included medical history, physical examination, and laboratory and imaging investigations ordered and documented by the treating physicians. This led us to remove some important parameters, such as nutritional status assessment with body mass index, from our study because these parameters were not available in almost all of the patient files. A significant number of patients were also excluded from the study because of multiple missing variables in their workups.

## Conclusion

In conclusion, in-hospital mortality in cirrhotic patients is high in Ethiopia. West Haven grade III or IV hepatic encephalopathy, History of previous admission within one year period, AKI, HCC, and MELD-Na Score, were found to be predictors of in-hospital mortality. Prompt identification and management of hepatic encephalopathy and its precipitant at an earlier stage is crucial. Routine screening for HCC in patients with cirrhosis is also important for diagnosing and treating the disease at an earlier stage. Patients with a history of admission within a year, AKI, and high MELD Na score also need closer follow-up for a better treatment outcome and survival.

AcknowledgmentWe would like to extend our sincere gratitude to the management and staff of Adera medical and surgical center, St. Paul's Hospital Millennium Medical College, and Tikur Anbessa Specialized Hospital for their support and cooperation throughout the duration of this study.

## Author contributions

**Conceptualization:** Tamrat Elias, Abate Bane Shewaye, Henok Fisseha, Abdulsemed Mohammed Nur.

**Data curation:** Tamrat Elias, Kaleb Assefa Berhane, Kibrab Bulto Kumsa, Biruck Mohammed Seid.

**Formal analysis:** Tamrat Elias, Kaleb Assefa Berhane, Asteray Tsige Minyilshewa.

**Investigation:** Tamrat Elias, Abate Bane Shewaye, Henok Fisseha, Abdulsemed Mohammed Nur, Kaleb Assefa Berhane, Asteray Tsige Minyilshewa, Kibrab Bulto Kumsa, Biruck Mohammed Seid.

**Methodology:** Tamrat Elias, Abate Bane Shewaye, Henok Fisseha, Abdulsemed Mohammed Nur, Asteray Tsige Minyilshewa.

**Project administration:** Tamrat Elias, Abate Bane Shewaye, Henok Fisseha, Abdulsemed Mohammed Nur, Kaleb Assefa Berhane, Asteray Tsige Minyilshewa, Kibrab Bulto Kumsa, Biruck Mohammed Seid.

**Resources:** Kibrab Bulto Kumsa, Biruck Mohammed Seid.

**Supervision:** Tamrat Elias, Henok Fisseha, Abdulsemed Mohammed Nur, Kaleb Assefa Berhane, Asteray Tsige Minyilshewa.

**Validation:** Tamrat Elias, Henok Fisseha, Asteray Tsige Minyilshewa, Kibrab Bulto Kumsa, Biruck Mohammed Seid.

**Writing – original draft:** Tamrat Elias, Kaleb Assefa Berhane, Kibrab Bulto Kumsa, Biruck Mohammed Seid.

**Writing – review & editing:** Abate Bane Shewaye, Henok Fisseha, Abdulsemed Mohammed Nur, Asteray Tsige Minyilshewa.

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
