## [Decision Letter · Decision Letter 0]

3 Jan 2025

PONE-D-24-44329Predictors of In-hospital Mortality among Cirrhotic Patients in Ethiopia: A Multicenter Retrospective StudyPLOS ONE

Dear Dr. Elias,

Thank you for submitting your manuscript to PLOS ONE. After careful consideration, we feel that it has merit but does not fully meet PLOS ONE’s publication criteria as it currently stands. Therefore, we invite you to submit a revised version of the manuscript that addresses the points raised during the review process.

We look forward to receiving your revised manuscript.

Kind regards,

Alessandro Granito

Academic Editor

PLOS ONE

Journal Requirements:

Reviewers' comments:

Reviewer's Responses to Questions

**Comments to the Author**

1. Is the manuscript technically sound, and do the data support the conclusions?

Reviewer #1: Yes

Reviewer #2: Yes

Reviewer #3: Yes

Reviewer #4: Partly

2. Has the statistical analysis been performed appropriately and rigorously? 

Reviewer #1: Yes

Reviewer #2: Yes

Reviewer #3: Yes

Reviewer #4: Yes

3. Have the authors made all data underlying the findings in their manuscript fully available?

Reviewer #1: Yes

Reviewer #2: Yes

Reviewer #3: Yes

Reviewer #4: Yes

4. Is the manuscript presented in an intelligible fashion and written in standard English?

Reviewer #1: Yes

Reviewer #2: Yes

Reviewer #3: No

Reviewer #4: Yes

5. Review Comments to the Author

Reviewer #1: This is to determine predictors of In-hospital Mortality among Cirrhotic Patients in Ethiopia, a developing country. The manuscript is well written

There are few issues the authors have to addressed

1. This could be because of the high viral hepatitis burden (second paragraph the reason for younger age of diagnosing liver cirrhosis..................... not only high burden but early age of acquisition of these viruses

2. The higher prevalence of liver cirrhosis and its complications in men is likely due to a combination of behavioral factors such as alcohol consumption, and biological factors such as lower levels of estrogen, which have been shown to have anti-fibrotic and anti-inflammatory effects on the liver that collectively increase their susceptibility to the development and progression of various liver diseases leading to cirrhosis ...................... this sentence is too long and therefore not clear

3. Kindly compare and contrast the percentage of ascites recorded in your study in the discussion section

4. Am not seeing authors contribution in the manuscript

5. Kindly check and rectify some of the reference

Reviewer #2: Thank you, dear authors, for this interesting manuscript. Here are my comments:

#Introduction: At the end of the introduction, please explicitly state that this manuscript has been reported in accordance with the STROBE guidelines. Cite the relevant reference and include the checklist at the end of the manuscript.

#Kindly provide the reference for the diagnostic criteria of cirrhosis mentioned in the study

#The manuscript requires minor grammatical corrections throughout to enhance clarity and readability.

Reviewer #3: Cirrhosis is a common disease, and as a multicenter study, 299 cases is not a lot actually. There have been many previous studies on the etiology, but the correlation analysis of related factors for death in patients with liver cirrhosis in Ethiopia is relatively few, so it has certain novelty and clinical value. This study calls attention to certain risk factors and take earlier prevention measures.

However, there are many defects in the manuscript that need to be modified:

1.Professional language polishing is needed. There are too many inappropriate spellings (including the Abstract) in the whole article, such as the first letter of many words in the sentence should be lowercase; abbreviations and full names of terms cross over and over again in the article; incorrect use of symbols.

2.In the paragraph 3 of Introduction, autoimmune liver disease should be also an important cause.

3. In the Methodology, the diagnostic criteria for the compensatory and decompensated stage of cirrhosis are quite different and should be distinguished.

4. The Child-Pugh and MELD Na scoring criteria need to be briefly mentioned in the Methodology.

5. The abbreviations of the terms in the Table need to be annotated.

6. What does COR refer to in Table2?

7. More discussion of hepatic encephalopathy is recommended.

8. Measures to prevent risk factors for death might be appropriately mentioned in the Discussion.

Reviewer #4: Section Operational definition

1. Line 124:

Liver cirrhosis: was diagnosed based on the presence of two or more of the following

It would be more appropriate to write

Liver cirrhosis was diagnosed based on the presence of two or more of the following :

2. Section Conclusion

Do you think that a study carried out in 3 hospitals out of several others is enough to conclude about the whole country?

6. PLOS authors have the option to publish the peer review history of their article (what does this mean? ). If published, this will include your full peer review and any attached files.

**Do you want your identity to be public for this peer review?** For information about this choice, including consent withdrawal, please see our Privacy Policy .

Reviewer #1: **Yes: ** Dr. Amoako Duah

Reviewer #2: No

Reviewer #3: No

Reviewer #4: No

---

## [Author Response · Author response to Decision Letter 1]

12 Feb 2025

PONE-D-24-44329

Predictors of In-hospital Mortality among Cirrhotic Patients in Ethiopia: A Multicenter Retrospective Study

Dear Editor and Reviewers;

I hope this letter finds you well. I would like to express my gratitude for taking the time to review our manuscript. Your constructive feedback and comments have been invaluable in improving the quality and clarity of our work.

I would like to address the concerns and comments raised in your review. I have carefully considered each one of them and have made the necessary changes to the manuscript accordingly.

So please find the following comments and responses.

Journal Requirements:

- I have revised the manuscript according to PLOS ONE’s style requirements.

Reviewers' comments:

Reviewer's Responses to Questions

Comments to the Author

1. Is the manuscript technically sound, and do the data support the conclusions?

Reviewer #1: Yes

Reviewer #2: Yes

Reviewer #3: Yes

Reviewer #4: Partly

2. Has the statistical analysis been performed appropriately and rigorously?

Reviewer #1: Yes

Reviewer #2: Yes

Reviewer #3: Yes

Reviewer #4: Yes

3. Have the authors made all data underlying the findings in their manuscript fully available?

Reviewer #1: Yes

Reviewer #2: Yes

Reviewer #3: Yes

Reviewer #4: Yes

4. Is the manuscript presented in an intelligible fashion and written in standard English?

Reviewer #1: Yes

Reviewer #2: Yes

Reviewer #3: No

Reviewer #4: Yes

- Thank you for the positive response. Some sections of the manuscript are now edited to make it clearer.

5. Review Comments to the Author

Reviewer #1: This is to determine predictors of In-hospital Mortality among Cirrhotic Patients in Ethiopia, a developing country. The manuscript is well written

- Thank you for the positive response.

There are few issues the authors have to address

1. This could be because of the high viral hepatitis burden (second paragraph the reason for younger age of diagnosing liver cirrhosis..................... not only high burden but early age of acquisition of these viruses

- Thanks for the insight. It is now included in the manuscript.

2. The higher prevalence of liver cirrhosis and its complications in men is likely due to a combination of behavioral factors such as alcohol consumption, and biological factors such as lower levels of estrogen, which have been shown to have anti-fibrotic and anti-inflammatory effects on the liver that collectively increase their susceptibility to the development and progression of various liver diseases leading to cirrhosis ...................... this sentence is too long and therefore not clear

- The sentence is now rewritten in a clear & concise manner.

3. Kindly compare and contrast the percentage of ascites recorded in your study in the discussion section

- The comparison of the prevalence of ascites in our study population with previous studies is now incorporated in the manuscript.

4. Am not seeing authors contribution in the manuscript

- Sorry for that. Authors contribution is now included in the revised manuscript.

5. Kindly check and rectify some of the reference

- Thanks for the feedback. The reference section is now edited.

Reviewer #2: Thank you, dear authors, for this interesting manuscript.

- Thank you for taking your time to review our manuscript.

Here are my comments:

#Introduction: At the end of the introduction, please explicitly state that this manuscript has been reported in accordance with the STROBE guidelines. Cite the relevant reference and include the checklist at the end of the manuscript.

- Statement of Reporting Compliance with the STROBE guidelines along with the reference is included at the end of the introduction. The checklist is also included with manuscript submission as a supplementary material.

#Kindly provide the reference for the diagnostic criteria of cirrhosis mentioned in the study

- The reference for the diagnostic criteria of cirrhosis used in the study is included in the revised manuscript.

#The manuscript requires minor grammatical corrections throughout to enhance clarity and readability.

- Thank you for the feedback. The manuscript is now edited for language usage, spelling, and grammar.

Reviewer #3: Cirrhosis is a common disease, and as a multicenter study, 299 cases is not a lot actually. There have been many previous studies on the etiology, but the correlation analysis of related factors for death in patients with liver cirrhosis in Ethiopia is relatively few, so it has certain novelty and clinical value. This study calls attention to certain risk factors and take earlier prevention measures.

- Thank you for the positive response.

However, there are many defects in the manuscript that need to be modified:

1.Professional language polishing is needed. There are too many inappropriate spellings (including the Abstract) in the whole article, such as the first letter of many words in the sentence should be lowercase; abbreviations and full names of terms cross over and over again in the article; incorrect use of symbols.

- The manuscript is now edited for language usage, spelling, and grammar.

2.In the paragraph 3 of Introduction, autoimmune liver disease should be also an important cause.

- Thank you for the reminder. Autoimmune hepatitis is now included as one of the major causes of liver disease.

3. In the Methodology, the diagnostic criteria for the compensatory and decompensated stage of cirrhosis are quite different and should be distinguished.

- We totally agree with the reviewer's comment and have included information about liver decompensation in the second paragraph of the introduction. The reason we didn’t include it in the methodology section is because the current study included all patients with cirrhosis despite their compensation status.

4. The Child-Pugh and MELD Na scoring criteria need to be briefly mentioned in the Methodology.

- The Child-Pugh and MELD Na scoring criteria are briefly discussed in the methodology (data collection tools and procedures) section of the revised manuscript.

5. The abbreviations of the terms in the Table need to be annotated.

- Thank you for your suggestion. A list of abbreviations with their full forms has been added at the base of the table to enhance clarity and readability.

6. What does COR refer to in Table2?

- COR in table 2 refers to crude odds ratio. Now, the abbreviation with its full form is included at the base of the table.

7. More discussion of hepatic encephalopathy is recommended.

- Thank you for the input. In the revised manuscript, discussion about the spectrum, prevalence, and classification of hepatic encephalopathy is included.

8. Measures to prevent risk factors for death might be appropriately mentioned in the Discussion.

- Measures to prevent risk factors for in-hospital mortality has now been included in the discussion section of the revised manuscript.

Reviewer #4: Section Operational definition

1. Line 124:

Liver cirrhosis: was diagnosed based on the presence of two or more of the following

It would be more appropriate to write

Liver cirrhosis was diagnosed based on the presence of two or more of the following :

- Thank you for the suggestion. Correction has been made accordingly.

2. Section Conclusion

Do you think that a study carried out in 3 hospitals out of several others is enough to conclude about the whole country?

- We acknowledge that including additional hospitals could enhance the generalizability of the findings, but the three hospitals included in the study are the largest referral centers in the country, which receive a high volume of patients with cirrhosis from various regions, and we believe the findings are representative of the broader national population.

---

## [Decision Letter · Decision Letter 1]

12 Mar 2025

PONE-D-24-44329R1Predictors of in-hospital mortality among cirrhotic patients in Ethiopia: A multicenter retrospective studyPLOS ONE

Dear Dr. Elias,

Thank you for submitting your manuscript to PLOS ONE. After careful consideration, we feel that it has merit but does not fully meet PLOS ONE’s publication criteria as it currently stands. Therefore, we invite you to submit a revised version of the manuscript that addresses the points raised during the review process.

We look forward to receiving your revised manuscript.

Kind regards,

Alessandro Granito

Academic Editor

PLOS ONE

Journal Requirements:

Reviewers' comments:

Reviewer's Responses to Questions

**Comments to the Author**

1. If the authors have adequately addressed your comments raised in a previous round of review and you feel that this manuscript is now acceptable for publication, you may indicate that here to bypass the “Comments to the Author” section, enter your conflict of interest statement in the “Confidential to Editor” section, and submit your "Accept" recommendation.

Reviewer #1: All comments have been addressed

Reviewer #3: (No Response)

2. Is the manuscript technically sound, and do the data support the conclusions?

Reviewer #1: Yes

Reviewer #3: Yes

3. Has the statistical analysis been performed appropriately and rigorously? 

Reviewer #1: Yes

Reviewer #3: Yes

4. Have the authors made all data underlying the findings in their manuscript fully available?

Reviewer #1: Yes

Reviewer #3: Yes

5. Is the manuscript presented in an intelligible fashion and written in standard English?

Reviewer #1: Yes

Reviewer #3: Yes

6. Review Comments to the Author

Reviewer #1: All my previous recommendations have been addressed. However, the first paragraph of the results section should be under the methods section, maybe at the sample size. the reasons should be giving for those not meeting the inclusion criteria.

Reviewer #3: 1.Please highlight or red the modified areas, otherwise it will increase the difficulty of reviewing again.

2.Although the language is much better than before, there are still some minor flaws, such as Child-Pugh is sometimes capitalized and sometimes lowercase, Median Age in Years maybe “Median age in years”

3.What COR is in Table 2 remains unexplained.

4. Please explain why cirrhosis cannot be distinguished and analyzed according to decompensated and decompensated periods?

7. PLOS authors have the option to publish the peer review history of their article (what does this mean? ). If published, this will include your full peer review and any attached files.

**Do you want your identity to be public for this peer review?** For information about this choice, including consent withdrawal, please see our Privacy Policy .

Reviewer #1: No

Reviewer #3: No

---

## [Author Response · Author response to Decision Letter 2]

19 Mar 2025

PONE-D-24-44329R1

Predictors of In-hospital Mortality among Cirrhotic Patients in Ethiopia: A Multicenter Retrospective Study

Dear Editor and Reviewers;

I hope this letter finds you well. I would like to express my gratitude for taking the time to review our manuscript. Your constructive feedback and comments have been invaluable in improving the quality and clarity of our work.

I would like to address the concerns and comments raised in your review. I have carefully considered each one of them and have made the necessary changes to the manuscript accordingly.

So please find the following comments and responses.

Journal Requirements:

- I have reviewed the reference list and found it to be complete, and no retracted paper was cited.

Reviewers' comments:

Reviewer's Responses to Questions

Comments to the Author

1. If the authors have adequately addressed your comments raised in a previous round of review and you feel that this manuscript is now acceptable for publication, you may indicate that here to bypass the “Comments to the Author” section, enter your conflict of interest statement in the “Confidential to Editor” section, and submit your "Accept" recommendation.

Reviewer #1: All comments have been addressed

Reviewer #3: (No Response)

- Thank you for the positive response.

2. Is the manuscript technically sound, and do the data support the conclusions?

Reviewer #1: Yes

Reviewer #3: Yes

- Thank you for the positive response.

3. Has the statistical analysis been performed appropriately and rigorously?

Reviewer #1: Yes

Reviewer #3: Yes

- Thank you for the positive response.

4. Have the authors made all data underlying the findings in their manuscript fully available?

Reviewer #1: Yes

Reviewer #3: Yes

- Thank you for the positive response.

5. Is the manuscript presented in an intelligible fashion and written in standard English?

Reviewer #1: Yes

Reviewer #3: Yes

- Thank you for the positive response.

6. Review Comments to the Author

Reviewer #1: All my previous recommendations have been addressed. However, the first paragraph of the results section should be under the methods section, maybe at the sample size. The reasons should be giving for those not meeting the inclusion criteria.

- Thank you for the suggestion. The first part of the result section is now included in the sample size section of the methodology part, and the explanation for being excluded from the study are also explained.

Reviewer #3:

1. Please highlight or red the modified areas, otherwise it will increase the difficulty of reviewing again.

- I apologize for any trouble this may have caused. The marked-up copy of our manuscript that highlights changes made to the original version is uploaded as a separate file labeled 'Revised Manuscript with Track Changes,' and an unmarked version of our revised paper without tracked changes is uploaded as a separate file labeled 'Manuscript.'.

2. Although the language is much better than before, there are still some minor flaws, such as Child-Pugh is sometimes capitalized and sometimes lowercase, Median Age in Years maybe “Median age in years”

- Thank you for the feedback. The aforementioned capitalization mistakes are now corrected.

3. What COR is in Table 2 remains unexplained.

- COR and AOR are now described as crude odds ratio and adjusted odds ratio at the bottom of the table.

4. Please explain why cirrhosis cannot be distinguished and analyzed according to decompensated and decompensated periods?

- Thank you for raising the idea. Analyzing patients survival based on their compensation status is advisable. However, the vast majority of our patients were admitted due to upper GI bleeding, hepatic encephalopathy, or ascites which are the signs of hepatic decompensation.

---

## [Decision Letter · Decision Letter 2]

24 Mar 2025

Predictors of in-hospital mortality among cirrhotic patients in Ethiopia: A multicenter retrospective study

PONE-D-24-44329R2

Dear Dr. Elias,

We’re pleased to inform you that your manuscript has been judged scientifically suitable for publication and will be formally accepted for publication once it meets all outstanding technical requirements.

Kind regards,

Alessandro Granito

Academic Editor

PLOS ONE

Additional Editor Comments (optional):

Reviewers' comments:

Reviewer's Responses to Questions

**Comments to the Author**

1. If the authors have adequately addressed your comments raised in a previous round of review and you feel that this manuscript is now acceptable for publication, you may indicate that here to bypass the “Comments to the Author” section, enter your conflict of interest statement in the “Confidential to Editor” section, and submit your "Accept" recommendation.

Reviewer #1: All comments have been addressed

Reviewer #3: All comments have been addressed

2. Is the manuscript technically sound, and do the data support the conclusions?

Reviewer #1: Yes

Reviewer #3: Yes

3. Has the statistical analysis been performed appropriately and rigorously? 

Reviewer #1: Yes

Reviewer #3: Yes

4. Have the authors made all data underlying the findings in their manuscript fully available?

Reviewer #1: Yes

Reviewer #3: Yes

5. Is the manuscript presented in an intelligible fashion and written in standard English?

Reviewer #1: Yes

Reviewer #3: Yes

6. Review Comments to the Author

Reviewer #1: The authors have addressed all my previous concerns. I have no other comments for the authors to address.

Reviewer #3: (No Response)

7. PLOS authors have the option to publish the peer review history of their article (what does this mean? ). If published, this will include your full peer review and any attached files.

**Do you want your identity to be public for this peer review?** For information about this choice, including consent withdrawal, please see our Privacy Policy .

Reviewer #1: No

Reviewer #3: No

---

## [Editor Report · Acceptance letter]

PONE-D-24-44329R2

PLOS ONE

Dear Dr. Elias,

I'm pleased to inform you that your manuscript has been deemed suitable for publication in PLOS ONE. Congratulations! Your manuscript is now being handed over to our production team.

Kind regards,

on behalf of

Dr. Alessandro Granito

Academic Editor

PLOS ONE